# TURING COMPLETE TRANSFORMERS: TWO TRANSFORMERS ARE MORE POWERFUL THAN ONE

## ABSTRACT

This paper presents Find+Replace transformers, a family of multi-transformer architectures that can provably do things no single transformer can, and which outperform GPT-4 on several challenging tasks. We first establish that traditional transformers and similar architectures are not Turing complete, while Find+Replace transformers are. Using this fact, we show how arbitrary programs can be compiled to Find+Replace transformers, aiding interpretability research. We also demonstrate the superior performance of Find+Replace transformers over GPT-4 on a set of composition challenge problems (solving problems 20x longer on tower of Hanoi, $3\% \rightarrow 100\%$ on multiplication, $72\% \rightarrow 100\%$ on a dynamic programming problem). This work aims to provide a theoretical basis for multi-agent architectures, and to encourage their exploration.

## 1 INTRODUCTION

The first computers – including the difference engine , differential analyzer, Z1, and ABC (Babbage & Babbage, 1825; Bush, 1931; Rojas, 2014; Atanasoff, 1940) – were not Turing Complete. Some such machines, like the Hollerith Tabulating Machine (Hollerith, 1889) and the Harvard Mark I (Comrie, 1946), even achieved considerable real-world use despite that limitation. However, the advent of Turing Complete computers (Turing et al., 1936; Goldstine & Goldstine, 1946; Kilburn, 1949) fundamentally changed how computers were used and led to the development of more complex, comprehensible, and composable programs (Backus, 1954; Copeland, 2004).

As we will show in this paper, current LLMs based on the transformer architecture (Vaswani et al., 2017) are not Turing Complete. We present an alternative that is.

The fundamental reason transformers are not Turing complete is that, once the architecture of a transformer is decided, there is a bounded amount of computation that it can do. This guarantees the model will fail to generalize beyond input of some length and complexity. Such limitations are not only theoretical, they are supported by a number of recent results on the ability of language models to generalize to large context lengths (Del'etang et al., 2022; Liu et al., 2023; Dziri et al., 2023).

Addressing these deficiencies is nontrivial and requires a fundamental shift in approach. We propose an approach drawing from multi-agent systems (Messing, 2003; Stone & Veloso, 2000), particularly multi-transformer systems. Such systems have recently garnered interest, being employed to generate simulacra of human behavior (Park et al., 2023), perform economic simulations (Horton, 2023), and demonstrate open-ended exploration in games like Minecraft (Wang et al., 2023a).

This paper presents a family of multi-transformer architectures, and provides theoretical and empirical evidence the family can outperform traditional transformers. We hope this study will ignite further investigations into architectures that are multi-transformer and Turing complete.

Our contributions are as follows:

- We provide a simple proof that current LLMs are not Turing Complete
- We present Find+Replace transformers, a family of provably Turing Complete architectures
- We introduce a method for turning any program into a Find+Replace transformer
- We show that Find+Replace transformers out-perform GPT-4 on a set of challenge tasks

## 2 TRANSFORMERS ARE NOT TURING COMPLETE

### 2.1 WHAT KINDS OF PROBLEMS ARE TRANSFORMERS UNABLE TO SOLVE BECAUSE THEY ARE NOT TURING COMPLETE?

It is unusual to use complexity theory to study transformers, so we feel that it is necessary to first explain why a complexity theoretic analysis of transformers is useful: the computational complexity of a model determines what kinds of problems the model can generalize to.

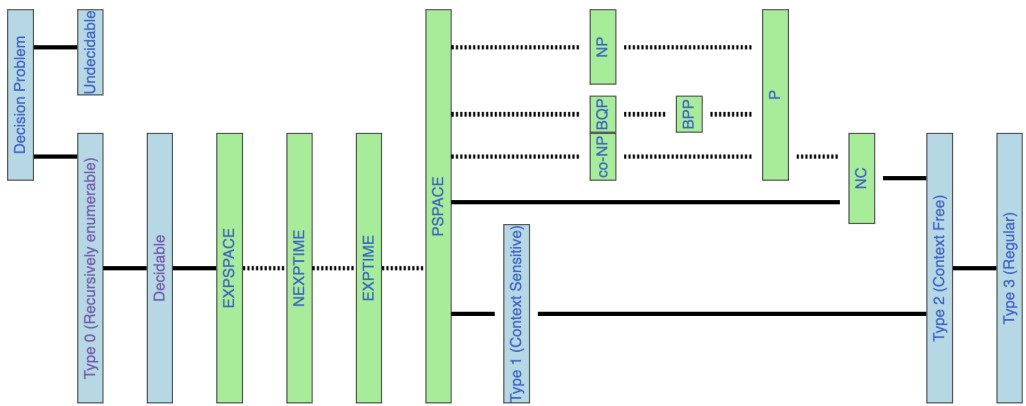

Figure 1: The above table (adapted from an online reference) shows some of the classes of problems that are considered in complexity theory. If class X is a strict subset of Y, then X is shown below Y with a dark line connecting them. If X is a subset, but it is unknown whether they are equal sets, then the line is lighter and dotted. The most computationally complex classes are shown on the left, and the least complex on the right.

In computational theory, complexity classes are used to categorize problems based on the resources required for their solution. These resources could be time (how many steps it takes to solve a problem), space (how much memory is needed), or other computational resources. A list of major complexity classes and their relationships can be found in Figure 1.

If a model can only learn algorithms in a lower class, it will be unable to generalize to examples in higher classes. Lookup tables provide a simple illustrative example.

What is the difference between a lookup table and a Turing Machine, and why do we prefer the latter? In theory, you could store the answer to arbitrarily many problems in a lookup table. The lookup table is even faster than the Turing Machine; it doesn't have to compute anything.

However, the inability to compute anything means that the lookup table cannot solve problems which it has not encountered before. Unless you have stored every possible case you care about, the lookup table *must* fail to generalize to new examples. A model with the computational complexity of a lookup table is only memorizing, not learning the underlying process of the task it was trained for.

This failure to generalize is not limited to lookup tables – any computational model which is not Turing Complete must fail to generalize to new examples of some problem. Problems higher up in the hierarchy of complexity classes cannot be solved by computational models which are lower in the hierarchy. If a model is lower in the hierarchy, it may be able to solve specific instances of the problem (by memorizing a solution to a simpler class of sub-problems); however, given a problem of a higher complexity above a certain size, the model will be unable to solve the problem.

Problems in the class REGULAR provide an illustrative example. They are processed by finite automata, which are computational models that have a finite number of states. They are equivalent to regular expressions. When processing a string, a finite automaton starts in a specific state, reads the characters in the string one by one, and transitions between states according to its transition function. The automaton accepts the string if it ends in an accepting state.

Let's consider a simple problem that finite automata cannot solve: recognizing the language $L = \{a^n b^n | n \geq 0\}$. This language consists of strings with $n$ 'a's followed by $n$ 'b's.

A regular expression or finite automaton can recognize strings in this language up to a certain length. For example, the regular expression $a^*b^*$ can recognize strings in $L$ up to length 2. However, no regular expression or finite automaton can recognize all strings in $L$.

This is because regular expressions and finite automata do not have the ability to count, which is necessary to ensure that the number of 'a's equals the number of 'b's. They can only remember a finite number of things (one for each state). While you can construct larger automata to solve larger instances of problem, you can never construct an automata which generalizes to all instances.

If your model can only learn finite automata, then it has fundamental limits to how well it can generalize – regardless of the size of the model or the input data provided.

Del'etang et al. (2022) explicitly test this hypothesis by training models on tasks from the Chomsky Hierarchy (Chomsky, 1956), consisting of Type 0-3 languages in Figure 1, and testing on examples larger than those in the training set. They find that transformers fail to generalize on tasks more complex than the class REGULAR.

What failures would we expect of models that are stateless in the same way as regular automata? Their states will branch either at special tokens or at special locations (the beginning and end of the output). This is precisely the behavior of large language models trained on large contexts: performance is highest when relevant information occurs at the beginning or end of the context, and significantly degrades when relevant information is the middle of long contexts (Liu et al., 2023).

Transformer models have also been shown to have difficulty in compositional tasks where solutions to sub-problems or simpler problems have to be combined to solve a larger problem (Dziri et al., 2023).

This behavior is also predicted by computational complexity theory – failures to generalize due to constraints in computational expressiveness will look like failures of composition.

Suppose we have a model that can only learn algorithms in P. This model would be able to solve many useful problems, such as sorting a list of numbers or finding the shortest path in a graph. However, it would not be able to solve NP-complete problems (unless P equals NP).

This limitation would manifest as a failure of composition. For example, consider the problem of finding a Hamiltonian cycle in a graph (a cycle that visits each vertex exactly once). This problem is NP-complete.

If we try to solve this problem by composing solutions to simpler problems that are in P, we will fail. For example, we might try to solve the problem by finding a cycle in the graph (which is in P) and then checking whether it is a Hamiltonian cycle (which is also in P). However, this approach does not work, because it requires the model to compose the answers for *all* possible cycles.

Therefore, a model that can only learn algorithms in P would fail to generalize to this problem. No matter how it composes solutions to simpler problems, it will always fail on some instances of the Hamiltonian cycle problem. And although it might solve simpler examples through memorization or other heuristics, this is a fundamental limitation of models that are not Turing complete.

**Definition 2.1.** A computational model is said to be Turing-complete if it can be used to simulate any Turing machine.

**Proposition 2.2.** *If a model is not Turing Complete, there must be some computation it cannot do.*

*Proof.* Let's say we have a model $m$ which is not Turing Complete. Because it is not Turing Complete, there exists some Turing Machine $t$ which it cannot implement. That Turing Machine $t$ does some computation. Therefore, there is a computation which $m$ cannot do. □

Only a Turing complete model can reliably generalize to the most complex problems.

## 2.2 A Proof That Transformers Are Not Turing Complete

### 2.2.1 Intuition

Before we jump into the details of the proof, we want to provide an intuition of why transformers are not Turing complete. Let's imagine a transformer which keeps generating output from the tokens it has previously produced. It takes some sequence of characters and outputs some sequence of characters, then repeats the process on the new sequence. These transformers have some fixed context length $k$ and some fixed vocabulary $v$. As such, there are at most $v^k$ unique sequences the model can take in as input, or produce. What happens when a transformer takes in an input it has seen before? Because transformers are stateless (as is required to train them in parallel on many sequences), it must produce the same output – resulting in an infinite loop. This implies that the computation we can perform is limited (either we produce at most $v^k$ sequences, or the transformer is trivially non-halting). In contrast, Turing machines can perform unlimited computation, as Turing highlighted in his groundbreaking paper (Turing et al., 1936).

We proceed to formalize this intuitive notion by developing a computational model which can be used to describe transformers.

### 2.2.2 A Model Of Computation

Language models read in sequences of language tokens as input and produce language tokens as output. Indeed, language modeling can be formalized as modeling $\underset{w_t}{\operatorname{argmax}} P(w_t|w_{t-k+1}, ..., w_{t-1})$, i.e. predicting the next word from a series of $k$ prior words. This requires the model to have some vocabulary of tokens $\Sigma$ from which we draw words $w$.

In this proof, we concern ourselves with a subset of language models, namely fixed-length sequence-to-sequence models.

**Definition 2.3.** Given a finite vocabulary of symbols $\Sigma$ and a sequence length $k$, a fixed-length sequence-to-sequence model $m$ is a function $m : \Sigma^k \to \Sigma^k$. We call the set of all fixed-length sequence-to-sequence models $\mathbb{M}_{FS}$.

In order to accurately model practical language models, we also make the following assumption:

**Assumption 2.4.** $\Sigma$ contains two special tokens, `<eos>` which represents the end of a sequence and `<pad>` which represents blank spaces in a sequence.

Later in this section, we will formally prove that the most common transformer architectures are in $\mathbb{M}_{FS}$. For now, we can intuitively see why $\mathbb{M}_{FS}$ provides a good way of representing transformers. The attention mechanism (Bahdanau et al., 2014) was originally proposed for machine translation, and the transformer architecture was originally applied to machine translation (Vaswani et al., 2017). As such, these models were formulated to take in an input sequence (the original language) and return an output sequence (the target language), learning a function from one to the other. When modeling language, we can predict the next token by taking in a sequence and outputting an updated sequence

$$w_{t-k+1}, ..., w_{t-1}$$
$$\mapsto w_{t-k+2}, ..., \underset{w_t}{\operatorname{argmax}} P(w_t|w_{t-k+1}, ..., w_{t-1})$$

Alternatively, we could output a completion, for example a completion with sequence length of $k = 5$:

$$As\ a\ language\ model\ I \mapsto have\ been\ trained\ to\ generate$$

With such models, we can do computation by the follow procedure:

**Definition 2.5.** Given an input sequence $x \in \Sigma^k$, we can run a model $m \in \mathbb{M}_{FS}$ to termination by repeatedly applying $m$ to $x$ until $m^n(x)$ contains an `<eos>` token. We call $n$ (alternatively, $n(m, x)$) the terminating number of $m$ on $x$.

Running a hypothetical model $m_{helpful}$ to completion on the following string:

$$As\ a\ language\ model\ I$$
$$\mapsto have\ been\ trained\ to\ generate$$
$$\mapsto responses\ that\ are\ intended\ to$$
$$\mapsto be\ helpful\ and\ informative\ \texttt{<eos>}$$

In the above case, the terminating number for $m_{helpful}$ on the sequence $x =$ "*As a language model I*", is 3.

Not all models will terminate on all sequences. Consider a model $m_{only\ a}$ which maps any sequence to a sequence consisting entirely of the symbol $a$. It never generates an $\texttt{<eos>}$ token, and therefore never terminates. In such cases, we say the terminating number $n(m, x) = \infty$.

Asking whether models in $\mathbb{M}_{FS}$ terminate is the equivalent question of asking whether Turing machines halt. Can we create an algorithm which decides whether $m \in \mathbb{M}_{FS}$ terminates?

The following is a well known theorem about Turing Machines:

**Theorem 2.6.** *The halting problem for Turing Machines is undecidable.*

Accordingly, the following is true:

**Lemma 2.7.** *If we can create an algorithm which reliably decides whether $m$ terminates for all $m \in \mathbb{M}_{FS}$, then no model in $\mathbb{M}_{FS}$ can be Turing Complete.*

*Proof.* Assume we can create an algorithm $H$ which reliably decides whether $m$ terminates for all $m \in \mathbb{M}_{FS}$. Further assume for contradiction that there exists a model $m \in \mathbb{M}_{FS}$ which is Turing Complete. By Definition 2.1, this means that $m$ can simulate any Turing Machine, including a Turing Machine that doesn't halt. However, by our assumption, $H$ can decide whether any model in $\mathbb{M}_{FS}$ terminates. This means we can decide whether $m$ terminates when it's simulating a non-halting Turing Machine, which contradicts Theorem 2.6. Therefore, our assumption that there exists a Turing Complete model in $\mathbb{M}_{FS}$ must be false. $\square$

**Theorem 2.8.** *No model in $\mathbb{M}_{FS}$ is Turing Complete.*

*Proof.* Every model $m \in \mathbb{M}_{FS}$ is parameterized by a vocabulary $\Sigma$ and a sequence length $k$.

Given a sequence $x$, consider the following algorithm:

1. Run $m$ for $|\Sigma|^k$ iterations on $x$.

2. If $\texttt{<eos>}$ exists in any of the sequences generated by running $m$, then return true (i.e. that $m$ terminates on $x$).

3. If $\texttt{<eos>}$ does not exist in any of the sequences generated by running $m$, then return false (i.e. that $m$ does not terminate on $x$).

There are only $|\Sigma|^k$ possible sequences of length $k$ over a vocabulary $\Sigma$. If $m$ has not terminated after $|\Sigma|^k$ steps, then by the pigeon hole principle, we have seem at least one sequence $x_*$ more than once. $x_*$ must be equal to $m^i(x)$ for some $i$.

Let's say we first saw $x_*$ at step $i$ and then again at step $j$ where $j > i$. $m^{j-i}(x_*) = x_*$, which means that we are in a loop: if we apply $m$ to $x_*$ $j - i$ times, we get back to $x_*$. Therefore, if $m$ has not terminated after $|\Sigma|^k$ steps, it will never terminate – we can decide whether any $m \in \mathbb{M}_{FS}$ terminates on any $x$.

By Lemma 2.7, this means that no model in $\mathbb{M}_{FS}$ can be Turing Complete. $\square$

### 2.2.3 AUTOREGRESSIVE MODELS

**Definition 2.9.** A model $m$ is autoregressive if it implements a function $f : \bigcup_{n<c} \Sigma^n \to \Sigma^{n+1}$, parameterized by some $c$, where $x \mapsto x \parallel s$ for some $s \in \Sigma$. Here, $\parallel$ denotes concatenation.

Using this construction, we can prove that a number of familiar models are not Turing complete.

**Lemma 2.10.** *Any autoregressive model $m$ must be in $\mathbb{M}_{FS}$*

*Proof.* Given an autoregressive model $m$, we can construct a model $m' \in \mathbb{M}_{FS}$ as follows:

1. Define $k = c$ and $\Sigma' = \Sigma \cup \{\texttt{<pad>}\}$.

2. For any input sequence $x \in \Sigma^n$ where $n < c$, pad $x$ with $c - n$ `<pad>` tokens to create a sequence $x' \in \Sigma'^k$.

3. Define $m' : \Sigma'^k \to \Sigma'^k$ such that $m'(x') = m(x) \parallel \texttt{<pad>}^{k-n-1}$.

It is clear that $m'$ is a fixed-length sequence-to-sequence model as per Definition 2.3, and thus $m' \in \mathbb{M}_{FS}$.

Furthermore, the value of $m$ can be read out of $m'$ by simply removing the `<pad>` tokens from the output of $m'$. This shows that any computation done by $m$ can also be done by $m'$, and thus any autoregressive model $m$ must be in $\mathbb{M}_{FS}$. ☐

An example of this construction can be seen by running an autoregressive transformer which enumerates the letters of the alphabet up to $e$ on the sequence $a\,b\,c$:

$$a\,b\,c\,<pad>\,<pad>$$
$$\mapsto a\,b\,c\,d\,<pad>$$
$$\mapsto a\,b\,c\,d\,e$$
$$\mapsto b\,c\,d\,e\,<eos>$$

Transformers – both decoder-only and encoder-decoder models – are auto-regressive (this is a well-known fact, but the curious reader can see Appendix B.1 for a proof), and accordingly are not Turing complete. A longer discussion of why models are not Turing complete, including a comparison to prior literature on the Turing completeness of transformer models can be found in Appendix A.1.

## 3 THE FIND+REPLACE TRANSFORMER

### 3.1 ARCHITECTURE

In Section 2.2, we establish that transformers are not Turing complete. Based on the proof, it appears that the reason transformers are not Turing complete has to do with the way in which they are used – in particular, autoregressively generating a sequence limits the amount of computation the model can do.

It has already been shown that allowing transformers to do more computation allows for better results (Wei et al., 2022; Wang et al., 2022; Yao et al., 2023). Recent work has augmented the ways in which we use models by giving them access to additional forms of memory and tools, creating AI 'agents' (Borgeaud et al., 2021; Bertsch et al., 2023; Modarressi et al., 2023; Schick et al., 2023; Wang et al., 2023b).

Agents appear to be a natural solution to the problem that transformers are not Turing Complete – by changing how we are using the models (giving them access to additional tools or memory) can we make them Turing complete?

**Find + Replace transformers** are multi-transformer systems that operate on a sequence of arbitrary length, which we call the *tape*. They are comprised of *Find Transformers*, *Replace Transformers*, and a *Map* which is a function from Replace transformers to ordered sets of Find Transformers.

**Find Transformers** identify parts of the sequence as input for Replace transformers. They each have a fixed context length $k$ and look at every $k$-length sub-sequence of the tape, selecting the particular sub-sequence which leads to the highest activation in the final layer.

**Replace Transformers** take in the sub-sequences identified by Find transformers as input. A replace transformer $r$ takes as input the concatenation of the sub-sequences identified by the find transformers $f \in Map(r)$, then outputs a sequence of length $k$. So the replace transformer $r$ has a context length $k + \sum_{f \in Map(r)} k_f$, where $k_f$ is the context length of the find head $f$. The output sequence of length $k$ then replaces the sequence found by the first find Find transformer in $Map(r)$.

## 3.2 Why Find+Replace Transformers Are Turing Complete

We can think of Find+Replace transformers as machines for learning *reductions*. Consider the lambda calculus (Church, 1936), the Turing complete system which forms the basis of functional programming languages. There are only three rules in this language, called reductions:

- Alpha Reduction: This is a renaming of bound variables. It's used to avoid naming conflicts. For example, in the lambda term $\lambda x.x$, we could alpha reduce it to $\lambda y.y$ without changing the meaning.

- Beta Reduction: This is the process of applying a function to its arguments. For example, in the lambda term $(\lambda x.x)y$ (which represents applying the function $\lambda x.x$ to the argument $y$), we could beta reduce it to $y$.

- Eta Reduction: This is a simplification of functions and arguments. If you have a function like $\lambda x.(fx)$, and $x$ does not appear in $f$, then this can be eta reduced to just $f$.

Each of these reductions is quite simple – the individual reductions can even be done statelessly and therefore implemented by a discrete finite automaton. Repeatedly applying such reductions, however, can be used to do any computation. The computational simplicity of the individual rules allows them to be learned efficiently, but their use in concert makes them Turing complete. Many Turing complete systems work by doing similar kinds of "find and replace" operations.

A formal proof proceeds in Appendix B.2 by using Find+Replace transformers to implement a specific set of reductions that have been shown to be Turing complete: tag systems (Post, 1943).

## 3.3 Programs & Find+Replace Transformers

Because Find+Replace transformers are Turing complete, they are capable of expressing any program or programming language. We demonstrate a practical example of this in Appendix C, where we show how any program in a low-level language implementing a Turing machine can be converted to a Find+Replace transformer. Because other languages (e.g. python) can be compiled into this language, it also allows the conversion of almost any program into a Find+Replace transformer. Alternatively, a language like llvm (which is already the compilation target of many languages) could be compiled into Find+Replace transformers.

The ability to turn transformers into programs has a number of potential uses. For example, transformers created in this way can encode priors about how practitioners think a problem should be solved, allowing for the initialization of non-random weights and faster convergence during training.

Lindner et al. (2023) similarly provide a method to convert programs into transformers, detailing how such a technique can be used in mechanistic interpretability research. By creating a model whose mechanisms are known, we can test interpretability techniques to determine whether they re-derive the original mechanism.

Such reductions can also be implemented in parallel and could benefit from many of the tools developed around programming. For example, large numbers of small transformers could implement changes in parallel over a sequence, orchestrated by systems similar to those used for efficient compilation of functional languages (see e.g. the G-Machine (Augustsson, 1984; Kieburtz, 1985)).

This gives us confidence that Turing complete and multi-agent architectures are beneficial not only because they improve the ability of models to generalize, but because they provide an opportunity for greater cross pollination between AI and traditional computer science.

Table 1: Largest Tower of Hanoi problems solved by each model. The problems start with all disks on the left, with the goal of moving all the disks to the right. The number of disks is the "problem size". A problem with size $n$ has a solution of length $2^n - 1$. F+R was tested until problems of size 7, then testing was stopped, as it could continue to solve problems indefinitely and the problem grows exponentially.

| Model Name | Problem Size | Solution Length |
|---|---|---|
| TEXT-DAVINCI-003 | 0 | 0 |
| GPT-3.5-TURBO | 0 | 0 |
| GPT-4 | 3 | 7 |
| F+R | **7*** | **127** |

## 4    EXPERIMENTS

In Sections 2 and 3, we provide theoretical evidence that the complexity class of a model determines its ability to generalize, that transformers are not Turing complete, and that Find+Replace transformers are. Accordingly, Find+Replace transformers should generalize better to difficult tasks on which existing state-of-the-art transformers fail. In this section, we run experiments to verify that this is true and to prove that Find+Replace transformers can still be trained efficiently despite being Turing complete.

For each task, we finetune several copies of a pre-trained 100M parameter model to serve as either find heads or replace heads. The models are finetuned on examples of each step of the task. When a find head is trained on a sequence, it is trained to provide an output of all 0s when run on a subsequence not containing relevant information and all 1s when run on a subsequence that should be passed to a replace transformer. Replace transformers are trained on the selected outputs of the find transformers and the subsequence they replace the replacable subsequence with. We compare to GPT-3, GPT-3.5-Turbo, and GPT-4.

### 4.1    TOWER OF HANOI

When GPT-4 (OpenAI, 2023) was released, researchers from Microsoft Research published a paper detailing a series of early experiments with GPT-4, titled *Sparks of Artificial General Intelligence* (Bubeck et al., 2023). In this paper, they also highlighted some of the limitations of this "early AGI", specifically homing in on *limitations of autoregressive architecture highlighted by GPT-4* (see Section 8 of that paper).

A problem they use to illustrate the limitations of the architecture is the Tower of Hanoi problem. It is an example of a complex reasoning problem, and the failure of GPT-4 to solve this problem was used to highlight the lack of planning that current transformers face when reasoning.

In Table 1, we compare the performance of several models in solving full Tower of Hanoi problems. The difficulty of these problems increases exponentially with size: problems of size $n$ have $2^n - 1$ steps in their solution. Find+Replace transformers out-perform all other models on this task, including generating correct solutions at least 18x longer than GPT-4 (at which point we terminated the experiment).

Tower of Hanoi is just one problem presented in Bubeck et al. (2023) with which GPT-4 struggles. However, many of the other problems presented there are difficult to evaluate; for example, the failure to generate poetry satisfying certain constraints. In order to show that the Find+Replace transformer beating GPT-4 is not a one-off occurence, but rather a reflection of its abilities to generalize, we evaluate its performance on additional reasoning problems.

### 4.2    SOLVING THE FAITH AND FATE TASKS

Dziri et al. (2023) provide a series of composition challenge problems in which large language models can solve simple versions of a problem, but fail to generalize to more complex forms of the same problem. The problems include Multiplication and Dynamic Programming.

Table 2: Performance of various models on the Faith and Fate tasks Dziri et al. (2023). In the case of multiplication, problem size indicates the number of digits in the multiplicands. In dynamic programming, it indicates sequence length. Fine-tuned models, few-shot models, and CoT models all see examples of size 2/3 in multiplication or 4/5 in dynamic programming and are evaluated on examples of the task with a greater size.

| | | MULTIPLICATION | | | | DYNAMIC PROGRAMMING | | | | |
|---|---|---|---|---|---|---|---|---|---|---|
| PROBLEM SIZE | | 2 | 3 | 4 | 5 | 4 | 5 | 6 | 7 | 8 |
| GPT-3 | ZERO-SHOT | 76 | 15 | 0 | 0 | 11 | 4 | 4 | 0 | 0 |
| | 4-SHOT | 82 | 18 | 0 | 0 | 33 | 18 | 10 | 4 | 0 |
| | CoT | 86 | 2 | 2 | 0 | 58 | 22 | 15 | 8 | 2 |
| | FINETUNED | 99 | 55 | 1 | 0 | 100 | 100 | 22 | 14 | 8 |
| GPT-4 | ZERO-SHOT | 99 | 59 | 4 | 0 | 58 | 43 | 36 | 28 | 12 |
| | 4-SHOT | 99 | 63 | 21 | 3 | 77 | 71 | 58 | 55 | 42 |
| | CoT | 99 | 68 | 25 | 3 | 94 | 91 | 88 | 84 | 72 |
| F+RT | FINETUNED | **100** | **100** | **100** | **100** | **100** | **100** | **100** | **100** | **100** |

Results of this experiment are shown in Table 2. On every task, as we increase the size of the problem, the Find+Replace transformer out performs the traditional transformers. This is not due to the fact that the Find+Replace transformer has been trained on this task – even the fined-tuned version of GPT-3 fails to generalize to large examples of complex composition problems. The difference in performance is particularly remarkable because the Find+Replace transformer is several orders of magnitude smaller than any of the transformers against which it is being compared. Accordingly, this improvement results from the higher computational class of the Find+Replace transformer.

## 5 CONCLUSION

In this paper, we prove that single transformers are not Turing complete, demonstrate that multi-transformer systems can be Turing complete by introducing the Find+Replace transformer (an example of such a Turing complete system), and use this system to out-perform state-of-the-art transformers like GPT-4. In this final section, we want to dwell on the implications of our results.

Wei et al. (2022) introduced Chain-of-Thought (CoT) prompting methods. The introduction of these methods allowed for a vast improvement in the reasoning abilities of large language models – leading to rapid progress on tasks such as math problem solving (Cobbe et al., 2021; Hendrycks et al., 2021), logical reasoning (Saha et al., 2018; Geva et al., 2021), and programming (Zelikman et al., 2022).

We can think of Chain-of-Thought prompting as a kind of algorithm for thought: produce a linear set of steps to reach a conclusion. Additional works have produced other such algorithms, for example self-consistency (Wang et al., 2022) or Tree-of-Thought prompting (Yao et al., 2023), which achieve better results by using more complex algorithms for thought. The computational complexity of a model determines the kind of algorithms it can learn. In order to have more complex algorithms for thought, we then need more computationally expressive models.

This becomes clearest when examining the most difficult problems – often we don't know whether they have solutions, let alone how long it will take to reach a solution. Such open-ended problems require open-ended exploration – open-ended computation. We cannot solve them with a bounded amount of time. Accordingly, unless a model is Turing complete, there are some algorithms for thought which it can never implement, and therefore some problems to which it can never generalize.

Single transformers are not Turing complete. Collections of transformers – AI agents – are.

We hope this provides a theoretical foundation for AI models capable of computing *any* function and therefore being vastly more intelligent, and encourages more work into multi-transformers systems and the algorithms they can implement.

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

## A  NOTES ON COMPUTATIONAL COMPLEXITY

### A.1  WHY AREN'T TRANSFORMERS TURING COMPLETE?

To some readers, the proposition that transformers are not Turing complete may be surprising. This is especially the case because several papers have previously claimed that transformers *are* Turing compete (Dehghani et al., 2018; Pérez et al., 2019; Bhattamishra et al., 2020; Pérez et al., 2021; Giannou et al., 2023). It is worth clarifying where our analysis differs from theirs.

Previous works show that *given a specific instance of a problem*, a transformer exists which can solve it. We show that there exist certain problems for which *no transformer can solve every specific instance*.

This difference arises from a difference in the assumptions used to analyze transformers – previous works assume a limit on the number of steps required to solve a problem (Pérez et al., 2019; Bhattamishra et al., 2020; Pérez et al., 2021) or the amount of memory used (Giannou et al., 2023; Dehghani et al., 2018). This constraint on the problem allows a transformer to be constructed which can solve it.

The way in which we conduct our analysis is also more typical in traditional computational theory – in theory we can make larger and larger discrete finite automata (DFA) to solve any particular instance of a problem; however, there is no DFA which generalizes to all problems and it is this limitation which relegates the languages recognizable by DFAs (the class REGULAR) to the lowest complexity class commonly studied (Figure 1). Turing complete models are needed for generalization. As we will show in Section 4, by constructing models which are Turing complete, we can generalize more effectively in real-world problems.

The result that transformers are not Turing complete is also surprising for another reason, however. Many systems are accidentally Turing complete, such as Conway's game of life (Gardener, 1970) or playing card games (Churchill et al., 2019). If Turing completeness emerges so readily in systems not intended to be Turing complete, is there a specific reason transformers are not Turing complete?

A potential explanation arises from the "parallelism tradeoff" introduced by Merrill & Sabharwal (2022). They show (assuming transformers with log-precision floats), that transformers implement algorithms in the complexity class $TC^0$ (the class of efficiently parallelizable algorithms). This complexity class is a subset of $P$ (unless $L = P$), meaning that even very basic algorithms could not be expressed by transformers. This tradeoff arises from the need for efficient training of models on large quantities of data, the very property which made transformers successful at language modeling. This problem is exacerbated by the way in which transformers are used (auto-regressive generation), both of which limit the computational expressiveness of transformers.

## B  ADDITIONAL PROOFS

### B.1  TRANSFORMERS ARE AUTO-REGRESSIVE, AND THEREFORE NOT TURING COMPLETE

A corollary of Theorem 2.8 is the following:

**Corollary B.1.** *Both decoder-only and encoder-decoder transformers are not Turing Complete.*

*Proof.* Let us start with decoder-only transformers.

The decoder-only transformer, introduced in Liu et al. (2018) and popularized by Radford & Narasimhan (2018) & Radford et al. (2019), is a language modeling architecture which implements a function using the decoder block from (Vaswani et al., 2017). They model sequence-transduction language modeling task $(m^1, ..., m^n) \mapsto (y^1, ..., y^\eta)$ as $w^1, ..., w^{n+\eta+1} = (m^1, ..., m^n, \delta, y^1, ..., y^\eta)$, where $\delta$ is a separator token. The key fact is that the tokens $y^i$ are generated autoregressively from the input tokens and the previous tokens, up to the max length $c = n + \eta$.

Because the decoder-only transformer is auto-regressive, by Lemma 2.10, the decoder-only transformer is in $\mathbb{M}_{FS}$. And by Theorem 2.8, it is not Turing Complete.

Now, let us proceed to the case of the encoder-decoder architecture (Vaswani et al., 2017).

In this architecture, a hidden state is first computed from the input sequence, then the hidden state is used to generate a series of autoregressively sampled tokens. The hidden state is generated by a series of identical transformer blocks. Each transformer block is a fixed-length sequence to sequence function. As such, the encoder can do a bounded amount of information.

Because the hidden state has a fixed size, it can be modeled as a single token at the beginning of the sequence. For example, if there are $h$ different possible hidden states which the model can output, then we can add $h$ tokens to the vocabulary of the model, and start the autoregressive decoding with an additional token representing the hidden state prepended to the input. As the decoder-only transformer is in $\mathbb{M}_{FS}$, the encoder-decoder transformer must also be in $\mathbb{M}_{FS}$. $\qquad\square$

### B.2    FIND+REPLACE TRANSFORMERS ARE TURING COMPLETE

**Definition B.2.** A tag system is a triple $(m, \Sigma, P)$ where $m$ is a positive integer called the deletion number, $\Sigma$ is a finite alphabet of symbols – all finite strings on $\Sigma$ (i.e. $\bigcup_{n\in\mathbb{N}}\Sigma^n$) are called words – and $P$ is a set of production rules $P : \Sigma \to \bigcup_{n\in\mathbb{N}}\Sigma^n$ that assigns a word (called a production) to each symbol in $\Sigma$.

To use tag systems for computation, we also assume the presence of a special symbol:

**Assumption B.3.** $\Sigma$ contains a special symbol, `<halt>` which represents the end of a sequence. A word is halting if it begins with `<halt>` or if it has length $< m$.

Tag systems implement transformations which take non-halting words to new words in the following manner

$$t : \bigcup_{n\in\mathbb{N}}\Sigma^n \to \bigcup_{n\in\mathbb{N}}\Sigma^n = x_1, ..., x_m||S \mapsto S||P(x_1)$$

That is to say, given a word, they produce a new word by removing the first $m$ symbols and appending the production of the first symbol.

We can use tag systems to do computation by iterating the transformation $t$, starting from some initial word and and halting when a halting word is produced.

Tag systems are useful to us because of a result from Minsky (1961):

**Theorem B.4.** *Any Turing machine can be represented as a tag system, i.e. tag systems are Turing complete.*

In particular, we will use the following result (Cocke & Minsky, 1964) to prove that Find + Replace transformers are Turing Complete:

**Theorem B.5.** *Any Turing machine can be represented by a tag system with deletion number $m = 2$. In particular, we can simulate a universal Turing machine with 2 symbols and $r$ states by a 2-tag system with $|\Sigma| = 17r$ symbols and productions of length at most 3.*

Our proof is as follows:

**Theorem B.6.** *Find+Replace Transformers are Turing Complete.*

*Proof.* We will proceed by proving that, given any 2-tag system with productions of length at most 3, we can be create a Find+Replace Transformer which implements that tag system.

Consider such a tag system $(m, \Sigma, P)$. Let us create a new vocabulary for our find and replace transformer: $\Gamma = \Sigma \bigcup\{< bos >, < eos >\}$. We can then represent any word $S$ of symbols in $\Sigma$ as a tape $S' =$`<bos>` $|| S ||$ `<eos>`.

We create 2 find heads, the first $f_b$ with a context length 2 and the second $f_e$ with context length 1. Then, we create two replace heads $r_d$ with output length 1 and $r_P$ with output length 4. We also define a $Map$ as follows

$$Map(r) = \begin{cases} \{(f_b)\} & \text{if } r = r_d \\ \{(f_e, f_b)\} & \text{if } r = r_P \end{cases}$$

Let $f_b$ always identify the `<bos>` token and the token immediately following. Let $f_e$ always identify the `<eos>` token. Let $r_d$ always return the $< bos >$ token. Let $r_P$ always return $P(f_b(S'))|| < eos >$.

Given a tape

$$< bos > ||x_1, ..., x_m||S|| < eos >$$

applying this Find+Replace transformer produces the string

$$< bos > ||S||P(x_1)|| < eos >$$

Which is precisely the transformation implemented by the original tag system, once the `<bos>` and `<eos>` tags are removed.

Applying Theorem B.5, we see that Find+Replace transformers can implement a universal Turing machine and are therefore Turing complete. □

## C    CONVERTING PROGRAMS TO FIND+REPLACE TRANSFORMERS

As an example of converting programs into Find+Replace transformers, we can use a simple Turing complete language. Take, for example, Brainfuck (Müller, 1993).

Brainfuck is an esoteric programming language created in 1993 by Urban Müller. It is known for its extreme minimalism, with only eight simple commands, a data pointer and an instruction pointer.

The eight commands of the Brainfuck language are:

- "+" : Increments the value at the current cell by one.
- "-" : Decrements the value at the current cell by one.
- "¿" : Moves the data pointer to the next cell (to the right).
- "¡" : Moves the data pointer to the previous cell (to the left).
- "." : Outputs the ASCII value at the current cell (i.e., prints the character on the screen).
- "," : Accepts one byte of input, storing its value in the current cell.
- "[" : If the value at the current cell is zero, skips to the corresponding "]" command. Otherwise, move to the next command.
- "]" : If the value at the current cell is zero, move to the next command. Otherwise, jump back to the corresponding "[" command.

Brainfuck is closely related to P", a language created by Corrado Böhm in 1964 to describe the smallest universal Turing machine. P" was explicitly based on the Turing machine. Accordingly, Brainfuck provides a simple example of how a Turing machine can be implemented through a series of reductions.

Such a system has already been implemented by Brändli (2017). They show how a Brainfuck interpreter can be implemented using a Find and Replace Regex.

We can convert the regex into a transformer by first converting it into a Discrete Finite Automaton (DFA), then auto-regressively taking in a sequence and maintaining the state of the DFA in the sequence while consuming the sequence.

Then, using the transformers created in this way, we can implement the Find and Replace operations, allowing us to implement a Brainfuck interpreter. Any program in brainfuck can then be executed on a Find+Replace transformers, and so too for any program in a language which can be compiled to Brainfuck.

This provides two distinct methods to turn transformers into programs: 1) to turn the underlying langauge into a Find+replace, and 2) to turn the operations implementing the program into a Find+Replace transformer.

