# OpenReview forum: "Turing Complete Transformers: Two Transformers Are More Powerful Than One"
_ICLR.cc/2024/Conference — Submitted to ICLR 2024_

### Official Review · Reviewer_erh8 · 2023-11-01

**Soundness:** 1 poor
**Presentation:** 1 poor
**Contribution:** 1 poor
**Rating:** 3
**Confidence:** 3

**Summary:**

This paper uses complexity theory to show that the original Transformers are not Turing complete. It then proposes a way to make Transformers Turing complete, called the Find-Replace Transformers. Experiments on several datasets show that Find-Replace Transformers can achieve good performance on composite tasks.

**Strengths:**

Easy to read.

**Weaknesses:**

I would say that this paper is more like a popular science article rather than an academic paper. First, the authors use a lot of space to explain what computational complexity is, why a lower complexity class (automata) cannot solve a problem of higher complexity (context-free language). These are very basic knowledge in computer science and everyone with a basic computer background is familiar with them. In particular, I do not think it deserves spending half a page demonstrating that $\\{0^n1^n:n\in\mathbb N\\}$ is not in the regular language.

Second, the authors do not present any related work in this area. Actually, studying Transformers using a computation perspective is common in the machine learning community and I can list a large class of works (see below). The authors did not cite or discuss any of the basic works in the community. The results in this paper are well-known given these prior works.

[1] On the turing completeness of modern neural network architectures. ICLR 2019.

[2] Saturated transformers are constant-depth threshold circuits.TACL, 2022.

[3] Transformers learn shortcuts to automata. ICLR 2023.

[4] Tighter bounds on the expressivity of transformer encoders. ICML 2023.

[5] Towards Revealing the Mystery behind Chain of Thought: A Theoretical Perspective. NeurIPS 2023.

Third, the conclusion in this paper is questionable and the proof is quite non-strict. It contradicts to [1], which shows that Transformers are Turing-complete. Similar results hold for autoregressive Transformers [5] (although it was not rigorously proved that autoregressive Transformers are Turing-complete). Moreover, in the submission the authors considered only Transformers with input and output of bounded lengths, which are quite strange since Turing machines do not pose constraints on the tape length. If the length is constrained in Transformers, they clearly do not match Turing machines.

**Questions:**

-

---

### Official Review · Reviewer_5hCM · 2023-11-02

**Soundness:** 2 fair
**Presentation:** 1 poor
**Contribution:** 2 fair
**Rating:** 3
**Confidence:** 3

**Summary:**

- The paper first shows that Transformers with a finite context length cannot be Turing complete.
- It then proposes Find+Replace Transformers:
    - The Find Transformer takes in a length-$k$ sequence, and marks tokens as important or not (e.g. by marking tokens with 1 or 0).
    - For the Replace Transformer, the input sequence is the concatenation of outputs of a set of Find Transformers, and the output is a sequence that should be used to replace a part of the input sequence.
- Find+Replace Transformers are shown to be Turing complete, since they can implement a tag system which is known to be Turing complete.

**Strengths:**

- The paper provides the necessary background knowledge.
- The paper shows impressive empirical results on Tower of Hanoi, multiplication, and dynamic programming.

**Weaknesses:**

- The finding that "Transformers with a finite context length is not Turing complete" has been known. Prior work's Turing completeness is based on infinite precision and infinite context length. Assuming a finite context length naturally makes the Transformer not Turing complete.
- The descriptions of the experiment are not detailed enough for reproduction; I'm not sure how the perfect results on multiplication and dynamic programming are achieved, and hence cannot verify its correctness.
- Many pieces of writing are overly assertive and inaccurate.
  - Sec 2.1: It's not accurate to say that "It is unusual to use complexity theory to study transformers", since there has been a significant body of work on connecting transformers and formal languages and complexity classes, some of which the paper has cited but didn't discuss properly.
  - Lemma 2.10: "any autoregressive model" is inaccurate: it should be "any autoregressive model with a finite context window. For example, the Turing completeness of RNNs has long been established.
  - In the discussion on related work in Sec A, the paper states that prior work use either limited number of steps or limited memory -- I'm not sure why this is true, since to my understanding, Perez et al. 2019 and Giannou et al. 2023 both rely on the Transformer being applied recursively and for unlimited times. Please let me know if I'm missing something.

- Missing related work: Selection-Inference: "selection" is similar to the Find Transformer, and "Inference" is similar to the Replace Transformer.
  - Selection-Inference: Exploiting Large Language Models for Interpretable Logical Reasoning. By Antonia Creswell, Murray Shanahan, Irina Higgins.
- Other minor writing points:
  - Definition 2.9: $\cup_{n< c} \Sigma^n \rightarrow \Sigma^{n+1}$ is problematic notation: $n$ is being quantified by $c$ and hence should not appear alone on the right hand side.
  - Section C: the two move data pointer operations don't compile properly. Please use the math environment.
  - Section 2 can be made much terser with details left to the appendix. There should be more details on the Find+Replace Transformer.

**Questions:**

- To make sure I'm understanding the 2-tag system idea correctly: Find+Replace Transformers are iteratively/repeated applied to apply the reduction rule for 2 symbols in the reduction to the 2-tag system. This repeated application is the key to making a finite context length system Turing complete. Is this understanding correct?
- How large should we think of $Map(r)$ in general (i.e. in addition to the tag system example in the appendix)?
- "Any program in brainfuck can then be executed on a Find+Replace transformer": even though brainfuck is minimalistic in terms of the vocabulary and compiler size, its execution can be long and exceeds the bounded context length of a Find+Replace transformer.

---

### Official Review · Reviewer_zEM4 · 2023-11-02

**Soundness:** 2 fair
**Presentation:** 2 fair
**Contribution:** 2 fair
**Rating:** 3
**Confidence:** 4

**Summary:**

The authors make a formal argument that the class of fixed-length seq-to-seq models is not Turing complete, present a new architecture -- Find and Replace Transformers -- that is Turing complete, and provide experiments indicating that the proposed architecture can be trained to solve difficult reasoning problems such as Towers of Hanoi and 4-digit multiplication.

**Strengths:**

The main strength of this paper lies in raising important questions about the transformer architecture, its Turing completeness (or lack thereof, under certain assumptions), and the need to explore alternative architectures beyond standard transformers. This is a fresh perspective, different from most papers working with transformers.

It also provides various background perspectives on such questions, which make for an interesting read.

Lastly, unlike recent theoretical attempts characterizing the representation power of transformers, this work also supports its proposal with experiments (but see concerns about it below).

**Weaknesses:**

I see 3 main weaknesses in the paper:

1. The paper is written in a somewhat hand-wavy way. The authors are perhaps trying to appeal to broad audience, but this leaves meany important technical details unclear, while spending precious space on generic background material, such as the 2 pages of section 2. I'll state two examples of this:

  *  **Example 1:**  The main proposal --- the find + replace architecture --- is never defined formally and rigorously in section 3. This is especially odd as the paper has at least 15 other formal definitions and claims. This makes it difficult to understand some of the technical details and appreciate the additional power such transformers are supposed to add.

  * E.g., what prevents a **single** transformer from simulating one step of a find + replace transformer by using finitely many heads (one for each Find transformer) and then stacking on top of it a Replace transformer? And if this is a valid simulation (which it appears to be) of a single step of a find + replace transformer, the added computational power must be coming from in-place replacement of k tokens in the input and iterating over this single step, rather than from the find + replace two-transformer architecture.

  * Clarity on this is especially important as otherwise the actual technical finding in the paper won't support even the title-level message or the opening sentence of the abstract --- it won't be the case that *two transformers are more powerful than one*, it would rather be the case that **sequential iteration + the ability to replace in-place** is what adds more power. In fact, this is perhaps not even surprising as the ability to make local edits to a tape and iterate is precisely what Turing machines rely on.

  * **Example 2:** The *experimental setup in section 4 hints that each find transformer (and replace transformer) is trained separately "on examples of each step of the task", but this is very unclear --- what are the "steps" of a task (such as multiplication)? How was data generated for each step? These are critical pieces to understand about the experimental setup in order to ensure that the empirical success is not a result of "hand coding" an algorithm in the proposed find + replace transformer training while simply doing end-to-end (i.e., basic input -> output; no algorithmic help) finetuning or even few-shot prompting for the baseline GPT models.

  * I would strongly suggest that the authors use the available space more for clarifying technical details and less for providing tutorial-style material on complexity theory etc.

2. Another weakness is that the authors don't clearly **place the result in the context of prior papers** that have already formally shown that transformer encoders are not Turing complete. E.g., the Merrill et al (2022) paper cited by the authors in Appendix A.1 already showed that such encoders are in TC^0, which is a subset of P and hence clearly not Turing complete. There is also other work by Angluin et al. and Cheng et al. on studying the representation limits of various formal models of transformers. The current work should be placed clearly in the context of these earlier formal analyses.

3. Lastly, the non-Turing-completeness results (Theorem 2.8, Cor B.1) make the assumption that models have a **finite context of size $k$** at each step. In fact, the proof relies critically on this assumption. But a justification of this is missing. While it's true that language models are typically trained on fixed context lengths, this is not really a limitation of the transformer architecture. In fact, in decoder-only and encoder-decoder models, even practical transformers get to causally attend to the *full* input and partial output so far, not just the last $k$ tokens.

  * First, this limitation of the model should be mentioned upfront in the abstract and introduction, not delayed till definition 2.3. Second, this raises the question of the relevance of the finding to practical transformers. At the least, the overall claim should be clarified to state that *finite-context* transformers aren't Turing complete.

**Questions:**

Please refer to the above "weaknesses" section for background on these questions:

1. What's the justification of allowing fixed-context lookup during decoding? (I understand that's how it's done when training a transformer, but generally not when decoding at evaluation time)

2. Is it correct that each step of your multi-transformer architecture can be simulated by a single transformer as above? If not, why?

3. How is the ability to "replace k tokens in-place" related to the chain-of-thought style generation? E.g., one way to simulate replacing k tokens in-place is to copy the entire input over as chain-of-thought and replace k tokens in the copy.

4. Can you provide more details of how exactly was your find + replace transformer was trained? What were the steps and how "hand-coded" (or not) was the algorithm to solve, say, the Towers of Hanoi or multiplication?

---

### Official Review · Reviewer_bz3o · 2023-11-05

**Soundness:** 2 fair
**Presentation:** 2 fair
**Contribution:** 2 fair
**Rating:** 3
**Confidence:** 4

**Summary:**

The paper proposes a transformer architecture inspired by lambda calculus and other known "Find and Replace" Turing-complete mechanisms on an infinite tape. The paper argues that existing LLMs are not Turing complete as they are finite automata, and propose adding an arbitrary length tape on which to operate. The experiments show that "Find-Replace" transformers indeed outperform GPT (even though this is theoretically guaranteed for large enough problems due to GPT's modest attention span).

**Strengths:**

As LLMs currently only use memory in form of a token stream consisting of the prompt and (so far) generated tokens, investigating new mechanisms that refer to external memory is certainly something that should be (and is) explored a lot more. The paper has nice exposition of proofs (although they come directly from existing theoretical CS).

**Weaknesses:**

The main claim that transformers with finite attention span are not computationally universal is both somewhat obvious and previously already stated, (e.g.Schurmaans, "Memory Augmented Large Language Models are Computationally Universal": https://arxiv.org/abs/2301.04589). The claim depends on the observation that existing architectures have a limited total token length. But, traditional computers also operate with finite memory, and so potentially more interesting question is what architectures, prompting and fine-tuning techniques are easier to "program." For example, the above referenced Schurmaan's paper also allows a transformer access to external memory, like this submission does. Jojic et al "GPT is becoming a Turing machine: Here are some ways to program it": https://arxiv.org/abs/2303.14310 deals with limited memory simply with "skip attention," where GPT is forced to ignore pieces of generated text delineated with special tags (though in that paper, skip attention just reduces the memory needs of their prompting technique to those of a usual computer which only needs the instructions and the latest state in the program to keep executing it).

Current LLMs have very limited token memory, but it is getting longer with each new version (not to mention all the work, which authors acknowledged, on LLMs within systems that allow additional API calls, including to external memory), and as I mentioned, every computer has finite memory anyhow. In that sense, strictly speaking, the author's suggested architecture is not practical as it requires an infinite tape. Then the interesting question is if the proposed architecture but with a finite tape (thus being in what authors refer to as M_{FS} and not Turing-complete) is better than other options that are also memory-limited. For example, the above mentioned Jojic et al paper shows solving DP problems with near perfect accuracy with GPT 3, demonstrating the importance of prompt design.

The experimental section needs to be flashed out more. It is not exactly clear how the Find and Replace transformers were tuned. What 100 million parameter model was tuned and on what?

**Questions:**

See the above. I'd love to understand experiments better. One could make Find and Replace transformers by hand to execute, say 2-tag system, which, btw, GPT can execute as well, until it runs out of tokens. The handcrafted Find/replace transformers would in fact work better than trained ones on any given task (but would then not be trained transformers but simply a way to express lambda calculus using transformers). One of the issues with trying to trigger LLMs into executing programs is that they were trained on lots of different data and it is not always totally predictable how they will react to a prompt instructing them to execute a program (see Jojic et al). I.e. if we want to reap the benefits of LLMs generalized, association-based, language "understanding" and synthesis with the ability to more strictly follow a computational mechanism, what are the tradeoffs? Can an LLM do both (act like ChatGPT to understand the intent, but also executing a program to search for an answer)?

---

### Author Response · Authors · 2023-11-23

We appreciate the feedback from reviewers, and will use it to improve any future versions of the paper. We agree that it is not currently in an optimal state to be published.

In writing the current iteration of this paper, we primarily solicited feedback from peers who were not as familiar with the literature on how computational complexity relates to machine learning. As a result, we have over-optimized for educating that audience, as opposed to stating our own results clearly and compellingly.

It appears to us, that there is a good deal of confusion about this particular sub-field of ML, which we'd like to highlight with two examples:

First, it's worth noting that different reviewers sometimes gave opposite critiques of the paper, e.g.
Reviewer erh8: The conclusion in this paper is questionable... It contradicts to [1], which shows that Transformers are Turing-complete
Reviewer bz3o: The main claim that transformers with finite attention span are not computationally universal is both somewhat obvious and previously already stated

Second, although many of the concepts we use are from basic CS, the original draft we wrote was repeatedly met by the question "but why does this matter?" -- and it was not until the basic CS concepts were explained that researchers in other ML subfields understood why the work was worth considering at all. Many of the details of the architecture were omitted to make room for the more general education about concepts covered in the paper.

It seems to us that many of these difficulties stem from differences in understanding of the basic CS concepts, for example what constitutes a reasonable model of computation or what the concept of "Turing Completeness" means and is intended to convey. How else are we supposed to land on a reasonable set of assumptions when different works present different assumptions?

We would appreciate any guidance from the reviewers on how these issues might be addressed in future versions of a paper, if they would be gracious enough to do so -- it would be a great help to us in better navigating this process. Although it is not clear to us how to address all the countervailing concerns in a way that would appease all readers, it is clear that rewriting is necessary to bring this paper into a sufficiently clear state for publication.

---

### Meta-Review · Area_Chair_fZ23 · 2023-12-04

**Metareview:**

Reviewers are unanimous in recommending rejecting the paper, and the authors themselves say "it is clear that rewriting is necessary to bring this paper into a sufficiently clear state for publication". The first result that transformers are not Turing complete is already known, and important prior work is not discussed; it is also not considered what happens when the context length (memory) becomes unbounded, in which case this first result no longer holds.
That being said, although the reviewers point out important prior work that is not discussed, some positives about the paper is that it gives a new architecture (find+replace transformers), and offers an easy to read overview of the basics of computational complexity.

**Justification For Why Not Higher Score:**

The paper is not written at the level of an ICLR paper, and is motivated primarily by the observation that transformers are not Turing complete (but that observation is no longer true in the infinite context-length limit)

**Justification For Why Not Lower Score:**

n/a

---

### Decision · Program_Chairs · 2024-01-16

Reject